EMBO
Molecular Medicine

# A normal genetic variation modulates synaptic MMP-9 protein levels and the severity of schizophrenia symptoms

Katarzyna Lepeta[1], Katarzyna J Purzycka[2,†], Katarzyna Pachulska-Wieczorek[2,†], Marina Mitjans[3], Martin Begemann[3], Behnam Vafadari[1], Krystian Bijata[4], Ryszard W Adamiak[2], Hannelore Ehrenreich[3,*], Magdalena Dziembowska[1,5,**] & Leszek Kaczmarek[1,***] (ID)

## Abstract

Matrix metalloproteinase 9 (MMP-9) has recently emerged as a molecule that contributes to pathological synaptic plasticity in schizophrenia, but explanation of the underlying mechanisms has been missing. In the present study, we performed a phenotype-based genetic association study (PGAS) in > 1,000 schizophrenia patients from the Göttingen Research Association for Schizophrenia (GRAS) data collection and found an association between the *MMP-9* rs20544 C/T single-nucleotide polymorphism (SNP) located in the 3′untranslated region (UTR) and the severity of a chronic delusional syndrome. In cultured neurons, the rs20544 SNP influenced synaptic MMP-9 activity and the morphology of dendritic spines. We demonstrated that Fragile X mental retardation protein (FMRP) bound the MMP-9 3′UTR. We also found dramatic changes in RNA structure folding and alterations in the affinity of FMRP for MMP-9 RNA, depending on the SNP variant. Finally, we observed greater sensitivity to psychosis-related locomotor hyperactivity in *Mmp-9* heterozygous mice. We propose a novel mechanism that involves MMP-9-dependent changes in dendritic spine morphology and the pathophysiology of schizophrenia, providing the first mechanistic insights into the way in which the single base change in the *MMP-9* gene (rs20544) influences gene function and results in phenotypic changes observed in schizophrenia patients.

**Keywords** dendritic spine morphology; Fragile X mental retardation protein; matrix metalloproteinase 9; phenotype-based genetic association study; single-nucleotide polymorphism

**Subject Categories** Chromatin, Epigenetics, Genomics & Functional Genomics; Genetics, Gene Therapy & Genetic Disease; Neuroscience

## Introduction

Matrix metalloproteinase 9 (MMP-9) plays a pivotal role in synaptic plasticity that underlies physiological processes and brain pathologies (Huntley, 2012; Vafadari *et al*, 2016). In response to glutamate activity, locally translated MMP-9 is further released from dendritic spines that harbor excitatory synapses (Wilczynski *et al*, 2008; Szepesi *et al*, 2013). Once activated, MMP-9 cleaves specific substrates, such as adhesion molecules and cell surface receptors (Vafadari *et al*, 2016), inducing downstream signaling and morphological changes. The morphology of dendritic spines correlates with their functional parameters (Arellano *et al*, 2007), and the aberrant morphology of dendritic spines is associated with multiple brain disorders (Lin & Koleske, 2010; Penzes *et al*, 2011). Alterations in synaptic MMP-9 abundance have been associated with numerous brain pathologies, including epilepsy, autism spectrum disorders, and alcohol and cocaine addiction (Levy *et al*, 2014; Lepeta & Kaczmarek, 2015; Vafadari *et al*, 2016).

Schizophrenia is a complex neuropsychiatric disorder that involves impairments in perception and cognition, as well as in avolition, culminating in a triad of positive, negative, and cognitive symptoms. Paramount features of the disease are perturbations in synaptic connectivity in the prefrontal cortex, impairments in *N*-methyl-D-aspartate (NMDA) signaling, and dendritic spine dysfunctions (Penzes *et al*, 2011; Glausier & Lewis, 2013). Notably,

1  Department of Molecular and Cellular Neurobiology, Nencki Institute of Experimental Biology, Polish Academy of Sciences, Warsaw, Poland
2  Department of RNA Structure and Function, Institute of Bioorganic Chemistry, Polish Academy of Sciences, Poznan, Poland‡
3  Clinical Neuroscience, Max Planck Institute of Experimental Medicine, DFG Research Center for Nanoscale Microscopy and Molecular Physiology of the Brain (CNMPB), Göttingen, Germany
4  Institute of Biochemistry and Biophysics, Polish Academy of Sciences, Laboratory of RNA Biology and Functional Genomics, Warsaw, Poland
5  Laboratory of Molecular Basis of Synaptic Plasticity, Centre of New Technologies, University of Warsaw, Warsaw, Poland
   *Corresponding author. Tel: +49 551 3899 628; E-mail: ehrenreich@em.mpg.de
   **Corresponding author. Tel: +48 22 5543 721; E-mail: m.dziembowska@cent.uw.edu.pl
   ***Corresponding author. Tel: +48 22 659 3001; E-mail: l.kaczmarek@nencki.gov.pl
   †These authors contributed equally to this work
   ‡Correction added on 1 August 2017 after first online publication: Affiliation 2 was corrected.

recent extensive genomewide association study (GWAS) identified the involvement of dendritic spines and synapse-related processes in schizophrenia but not in bipolar or major depressive disorder (Network and Pathway Analysis Subgroup of Psychiatric Genomics Consortium, 2015).

The well-established role of MMP-9 in regulating NMDA signaling, dendritic spine morphology, and prefrontal and hippocampal function (Huntley, 2012; Vafadari *et al*, 2016) has drawn attention to the potential involvement of MMP-9 in the development of schizophrenia (Lepeta & Kaczmarek, 2015). Domenici *et al* (2010) reported elevated plasma levels of MMP-9 and its endogenous inhibitor, tissue inhibitor of matrix metalloproteinases (TIMP-1), in schizophrenia patients (also see Chang *et al*, 2011; Yamamori *et al*, 2013). Moreover, two independent studies found association of MMP-9 promoter polymorphism −1562C/T (rs3918242) with schizophrenia (Rybakowski *et al*, 2009; Han *et al*, 2011). Furthermore, MMP-9 mRNA levels were strikingly upregulated in blood cells in treatment-naive schizophrenia patients to decline after antipsychotic treatment (Kumarasinghe *et al*, 2013). Moreover, the chromosome region 20q11-13, where the *MMP-9* gene is located (Jean *et al*, 1995), has been linked to schizophrenia (Gurling *et al*, 2001).

Recent studies have shown that the synaptic translation of MMP-9 is regulated by Fragile X mental retardation protein (FMRP) (Dziembowska *et al*, 2012; Janusz *et al*, 2013; Gkogkas *et al*, 2014). FMRP is an RNA binding protein that is known to translationally repress the pool of neuronal mRNAs (Darnell *et al*, 2001; Zalfa *et al*, 2003; Hou *et al*, 2006; Muddashetty *et al*, 2007). Missing FMRP leads to Fragile X syndrome (FXS) (Rudelli *et al*, 1985). FMRP has also been implicated in affective disorders, attention-deficit/hyperactivity disorder, bipolar disorder, and schizophrenia [reviewed in Bryant and Yazdani (2016)]. Interestingly, aberrations in dendritic spines that are observed in FXS patients (Rudelli *et al*, 1985) and *Fmr1* knockout mice (Comery *et al*, 1997) have been linked to elevated synaptic levels of MMP-9 (Bilousova *et al*, 2009; Dziembowska *et al*, 2013; Gkogkas *et al*, 2014; Sidhu *et al*, 2014).

In the present translational study, we report an association between the *MMP-9* rs20544 C/T single-nucleotide polymorphism (SNP) in the 3′untranslated region (UTR) with the severity of schizophrenia symptoms. The 3′UTR plays an essential role in mRNA transport and local translation, thus influencing protein abundance at synapses (Martin & Ephrussi, 2009). We hypothesized that the *MMP-9* rs20544 SNP may affect synaptic availability of the enzyme and mediate the observed phenotype. We found that the MMP-9_C variant produces lower MMP-9 activity at the synapse than MMP-9_T, paralleled by differences in the distribution of the dendritic spine shape. To elucidate the underlying mechanism, we identified dramatic changes in RNA structure folding and alterations in the affinity of FMRP for MMP-9 RNA that depended on the polymorphic variant.

# Results

### Case–control analysis does not reveal an influence of the *MMP-9* rs20544 SNP on the risk of schizophrenia

To first evaluate whether the *MMP-9* rs20544 SNP is a risk SNP for schizophrenia, a case–control analysis was performed ($n = 1,087$

schizophrenia patients vs. $n = 1,235$ healthy controls; Hardy–Weinberg $P > 0.05$). This analysis found that rs20544 did not contribute to a higher risk of schizophrenia in our population, demonstrated by genotypic and allelic $\chi^2$ comparisons (genotypic: $\chi^2 = 0.948$, df = 2, $P = 0.623$; allelic: $\chi^2 = 0.195$, df = 1, $P = 0.659$; Fig 1A).

### Schizophrenia patients who carry the *MMP-9* rs20544 CC/CT genotype display more severe chronic delusions

To search for potential behavioral consequences of *MMP-9* rs20544 variants, we conducted a phenotype-based genetic association study (PGAS) in schizophrenia patients from the Göttingen Research Association for Schizophrenia (GRAS) data collection (Begemann *et al*, 2010; Ribbe *et al*, 2010). PGAS revealed a chronic delusion syndrome as the best-suited target phenotype. The respective PANSS items taken together most adequately reflect the behavioral phenotype that is associated with the risk genotype (C allele). Therefore, a *chronic delusion composite score* was generated that enables quantification and is characterized by excellent internal consistency of its subitems (Cronbach's α = 0.833; Fig 1B). Operationalization of the *chronic delusion composite score* is explained in Materials and Methods. The *chronic delusion composite score* showed a significant genotype (rs20544) association. Individuals who carried the C allele (CC/CT) had a higher score than TT carriers ($P = 0.0003$; Fig 1C). The *PANSS* P1 item "delusions" made a major contribution to the score (Fig 1D) and could have been used as single readout. However, *PANSS* P1 does not distinguish between acute or chronic delusion. Considering the other (borderline) significant subitems of the *chronic delusion composite score* yields a better description of the clinical picture of risk allele carriers, which is associated with social and emotional withdrawal, somatic concern, unusual thought content, and preoccupation (Fig 1D). Basic sociodemographic information as well as general and schizophrenia-related clinical parameters did not show any differences between CC/CT (risk allele) and TT carriers (Table 1).

### rs20544 C/T polymorphism affects MMP-9 mRNA structure

Although T is replaced by U in mRNA, for simplification we refer to the *MMP-9* rs20544-T variant as MMP-9_T throughout the manuscript, regardless of whether it refers to the RNA or DNA sequence. Accordingly, *MMP-9* rs20544-C is referred to as MMP-9_C.

SNPs have the ability to significantly alter RNA secondary structures (Pachulska-Wieczorek *et al*, 2006). *In silico* ss-count analysis (i.e., the probability that a given chain of nucleotides is single-stranded) for the isolated human 3′UTR (192nt length) of MMP-9 mRNA demonstrated a high probability of structural changes in the region that hosted the SNP. Smaller changes were also observed elsewhere in the 3′UTR sequence. Overall, the tertiary structure of MMP-9 mRNA showed major structural differences that depended on the SNP under study (Fig EV1).

To experimentally validate the *in silico* structural models, RNA structure probing by selective 2′-hydroxyl acylation analyzed by primer extension (SHAPE) analysis revealed that 469nt at the 3′end of MMP-9 mRNA formed a structurally distinct domain, and this sequence fragment was used to further refine the models. Experimentally supported MMP-9_C and MMP-9_T RNA structures of 3′-terminal domains of MMP-9 mRNA that contained 3′UTR are shown

**A** **Case-control analysis**

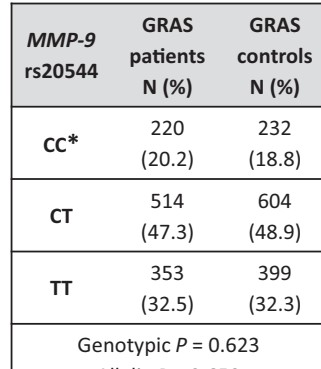

| *MMP-9* rs20544 | GRAS patients N (%) | GRAS controls N (%) |
|---|---|---|
| CC* | 220 (20.2) | 232 (18.8) |
| CT | 514 (47.3) | 604 (48.9) |
| TT | 353 (32.5) | 399 (32.3) |
| Genotypic *P* = 0.623 Allelic *P* = 0.659 | | |

*MAF: C = 0.435*

**B** *Chronic delusion composite score*

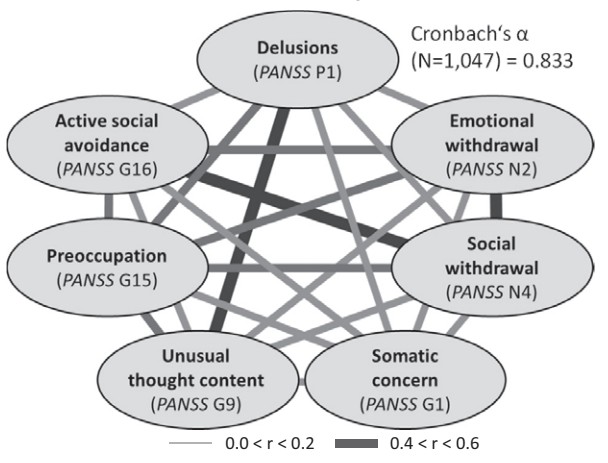

Cronbach's α (N=1,047) = 0.833

- Delusions (*PANSS* P1)
- Active social avoidance (*PANSS* G16)
- Emotional withdrawal (*PANSS* N2)
- Preoccupation (*PANSS* G15)
- Social withdrawal (*PANSS* N4)
- Unusual thought content (*PANSS* G9)
- Somatic concern (*PANSS* G1)

0.0 < r < 0.2    0.4 < r < 0.6
0.2 < r < 0.4    r > 0.6

**C** **Schizophrenia population (GRAS patients)**

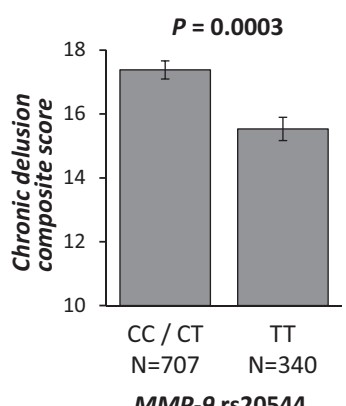

*P* = 0.0003

*Chronic delusion composite score*

CC / CT N=707     TT N=340

*MMP-9* rs20544

**D** *Chronic delusion composite score* – single items (GRAS patients)

| *MMP-9* rs20544 | Delusions | Emotional withdrawal | Social withdrawal | Somatic concern | Unusual thought content | Preoccupation | Active social avoidance |
|---|---|---|---|---|---|---|---|
| CC / CT (N= 711-716)* | 2.75±0.07 | 2.86±0.57 | 2.67±0.06 | 2.04±0.05 | 2.35±0.06 | 2.46±0.05 | 2.23±0.05 |
| TT (N=341-343)* | 2.34±0.09 | 2.59±0.08 | 2.45±0.08 | 1.81±0.06 | 2.11±0.08 | 2.27±0.07 | 1.98±0.07 |
| | *P* = 0.0005 | *P* = 0.012 | *P* = 0.069 | *P* = 0.049 | *P* = 0.022 | *P* = 0.075 | *P* = 0.06 |

*Sample sizes vary due to missing data on phenotyping*

**Figure 1.  Phenotype-based genetic association study (PGAS) exploring the impact of the *MMP-9* rs20544 polymorphism on the severity of chronic delusion.**

A  Case–control analysis reveals similar genotypic and allelic distributions of *MMP-9* rs20544 genotypes in patients and controls. MAF, minor allele frequency.

B  High intercorrelation of PANSS items that constitute the *chronic delusion composite score* (Spearman rank correlation coefficients), reflected by the high internal consistency of the scale (Cronbach's α = 0.833).

C  Genotype effect of *MMP-9* rs20544 on the *chronic delusion composite score*. C carriers (CC/CT) had a significantly higher score than TT carriers.

D  Association between rs20544 genotypes and subitems of *the chronic delusion composite score*.

Data information: The data are expressed as mean ± SEM and two-tailed *P*-values (Mann–Whitney *U*-test; data corrected by linear regression analysis for age, medication, and disease duration).

in Fig 2A. Although both structures contained three multihelical junctions (annotated E1, E2, and E3 in Fig 2A), the junctions were formed by interactions between different RNA sequences. Only E1 was similar for both RNAs. Interestingly, the SNP (nt 2147 in Fig 2A) in the MMP-9_C RNA is located within the hairpin stem, whereas MMP-9_T is present in the hairpin apical loop. Notably, even the folding of the distal fragments, such as the G-rich region (localized between nucleotides 2,258–2,271 of the MMP-9 mRNA 3′UTR) potentially recognized by FMRP, was influenced by the polymorphism (Fig 2A, blue boxes).

**FMRP binds to G-rich sequence in MMP-9 3′UTR**

MMP-9 3′UTR *in silico* sequence analysis (Kikin *et al*, 2006) revealed that the human MMP-9 3′UTR harbored three G-rich sequences that are putative FMRP binding sites. To identify the FMRP binding sites within the MMP-9 mRNA sequence, we performed RNA electrophoretic mobility shift assay (REMSA) with labeled MMP-9 RNA probes that spanned two predicted G-rich

sequences in either a wild-type form (wild-type probe) or a form that harbored either a mutation or deletion within one of the two G-rich sequences (mutant and delta probes, respectively; Fig 2B). Interestingly, we observed three shifted bands for the wild-type probe, suggesting the presence of three FMRP binding sites in the tested probe. One of the FMRP/MMP-9 RNA complex bands was clearly visible for the wild-type probe, and binding was hardly detectable for the probes that harbored a mutation or deletion (Fig 2C), indicating that FMRP interacts with this particular G-rich sequence.

**rs20544 C/T SNP affects the affinity of FMRP binding to MMP-9 mRNA**

Our RNA structure models revealed extensive structural changes in the MMP-9 mRNA molecule that depended on the rs20544 C/T polymorphism. Interestingly, one of the regions that was differentially folded for the two variants was the G-rich sequence, which may be implicated in the binding of FMRP (Fig 2A, blue boxes). The REMSA

**Table 1. Göttingen Research Association for Schizophrenia (GRAS) sample description.**

|  | Total GRAS sample (N = 924–1,059)[a] | CC/CT carriers (N = 625–716)[a] | TT carriers (N = 299–343)[a] | P value (Z/χ² value)[b] |
|---|---|---|---|---|
| Age, years[c] | 39.27 (12.49) [17.49 to 78.49] | 39.40 (12.62) [17.49 to 78.49] | 38.99 (12.22) [18.08 to 73.40] | $P = 0.754$ ($Z = -0.314$) |
| Male, No. (%) | 712 (67.2%) | 490 (68.4%) | 222 (64.7%) | $P = 0.228$ ($\chi^2 = 1.451$) |
| Education, years[c] | 12.16 (3.12) [0 to 27] | 12.14 (3.02) [0 to 23] | 12.19 (3.34) [8 to 27] | $P = 0.650$ ($Z = -0.454$) |
| Cognitive composite score[c,d] | −0.014 (0.85) [−2.57 to 2.98] | −0.023 (0.83) [−2.57 to 2.03] | 0.006 (0.87) [−2.30 to 2.98] | $P = 0.497$ ($Z = -0.679$) |
| Handedness, No. R,L,A[e] (%) | 944 (89.9%), 73 (7%), 33 (3.1%) | 634 (89.4%), 55 (7.8%), 20 (2.8%) | 310 (90.9%), 18 (5.3%), 13 (3.8%) | $P = 0.245$ ($\chi^2 = 2.812$) |
| Duration of disease, years[c] | 12.88 (10.58) [0.01 to 58.44] | 12.48 (10.48) [0.01 to 58.44] | 13.71 (10.77) [0.04 to 51.96] | $P = 0.054$ ($Z = -1.927$) |
| Chlorpromazine equivalents[c] | 687.40 (704.84) [0 to 7375] | 673.26 (649.17) [0 to 5728] | 716.96 (809.24) [0 to 7375] | $P = 0.854$ ($Z = -0.184$) |
| GAF score[c] | 45.89 (17.18) [2 to 90] | 45.38 (17.21) [5 to 90] | 46.98 (17.07) [2 to 90] | $P = 0.111$ ($Z = -1.592$) |
| CGI score[c] | 5.57 (1.07) [2 to 8] | 5.60 (1.07) [2 to 8] | 5.49 (1.07) [2 to 8] | $P = 0.070$ ($Z = -1.809$) |
| Current smokers, No. (%) | 726 (69.3%) | 491 (69.5%) | 235 (68.7%) | $P = 0.784$ ($\chi^2 = 0.075$) |
| No. of cigarettes/day[c] | 16.40 (14.81) [0 to 80] | 16.71 (15.32) [0 to 80] | 15.78 (13.70) [0 to 80] | $P = 0.804$ ($Z = -0.248$) |
| Current cannabis use, No. (%) | 127 (12.5%) | 87 (12.7%) | 40 (12.0%) | $P = 0.761$ ($\chi^2 = 0.092$) |
| Alcoholism severity score[f] | 0.024 (0.65) [−1.25 to 1.92] | 0.042 (0.66) [−0.95 to 1.92] | −0.013 (0.64) [−1.25 to 1.91] | $P = 0.240$ ($Z = -1.175$) |

GAF, Global Assessment of Functioning (American Psychiatric Association, 2000); CGI, Clinical Global Impression (Guy, 1976).
[a]Sample sizes vary due to missing data on phenotyping.
[b]Statistics of the comparison between CC/CT carriers and TT carriers; Mann–Whitney *U*- or chi-square test used.
[c]Mean (SD) [Range].
[d]Cognitive composite score consists of reasoning (Leistungsprüfsystem-subtest-3), executive function (Trail-Making Test B), verbal learning & memory test (VLMT) (Begemann *et al*, 2010; Stepniak *et al*, 2015).
[e]R = right-handers, L = left-handers, A = ambidexters.
[f]Alcoholism severity score integrates numbers of alcohol-related detoxifications, highest amount of regular drinking, frequency of drinking, number of positive SCID (Structured Clinical Interview for *DSM-IV* Disorders) items, and chronicity (Ribbe *et al*, 2011).

confirmed that FMRP bound to MMP-9 RNA (see above). Therefore, we tested whether the ability of FMRP to bind MMP-9 mRNA is different for the two polymorphic variants of MMP-9 mRNA molecules.

To maximize proper RNA structure folding, we performed a REMSA (Fig 3A) with probes that spanned 469 nt of the 3′-terminal residues of MMP-9 mRNA (Fig 3B). As shown in Fig 2A, these sequence fragments formed a structurally distinct domain. The incubation of MMP-9 probes with increasing concentrations of purified FMRP resulted in the formation of FMRP/MMP-9 RNA complexes. The pattern of RNA/protein complex migration in gel was characteristic of multiple binding sites (Fig 3A). These results corresponded well with our previous experiments with shorter ~80 nt probes (Fig 2C), for which three distinct complexes were observed, further confirming the presence of multiple FMRP binding sites within the MMP-9 mRNA.

Interestingly, we found significant changes in complex migration that depended on the SNP variant. The complex with MMP-9_C RNA migrated faster than the complex with MMP-9_T, compared with the probe alone (Fig 3C). For FMRP concentrations >800 nM, we did not observe significant differences in migration (Fig 3C, last column), possibly because of saturation of the RNA probe that bound the protein.

To assess the affinity of FMRP binding to MMP-9_C and MMP-9_T RNA probes, we performed a filter binding assay, indicating that FMRP bound more strongly to the MMP-9_C probe (Fig 3D) with the dissociation constant ($K_d$) for the two complexes: $K_d = 113.1 \pm 22.82$ nM for MMP-9_C, $K_d = 154.3 \pm 20.15$ nM for MMP-9_T probe.

**rs20544 C/T SNP influences MMP-9 activity at dendritic spines**

Fragile X mental retardation protein controls the local translation of MMP-9 mRNA at the synapse, and the lack of FMRP in *Fmr1* knockout mice leads to an increase in synaptic MMP-9 levels (Bilousova *et al*, 2009; Dziembowska *et al*, 2013; Janusz *et al*, 2013; Gkogkas *et al*, 2014). We investigated whether differences in FMRP binding affinity for rs20544 C/T polymorphic variants influence the synaptic availability of MMP-9. To measure synaptic MMP-9 activity, we used a DQ-gelatin assay with primary rat hippocampal neurons. Neurons were transfected with a plasmid that coded for the MMP-9_C or MMP-9_T variant and constitutively expressed red fluorescent protein (RFP) to visualize cell morphology. As a readout of MMP-9 activity, a relative increase in fluorescence was measured for individual spines in time-lapse imaging before, 15 and 40 min after stimulation. cLTP of synaptic activity was produced by

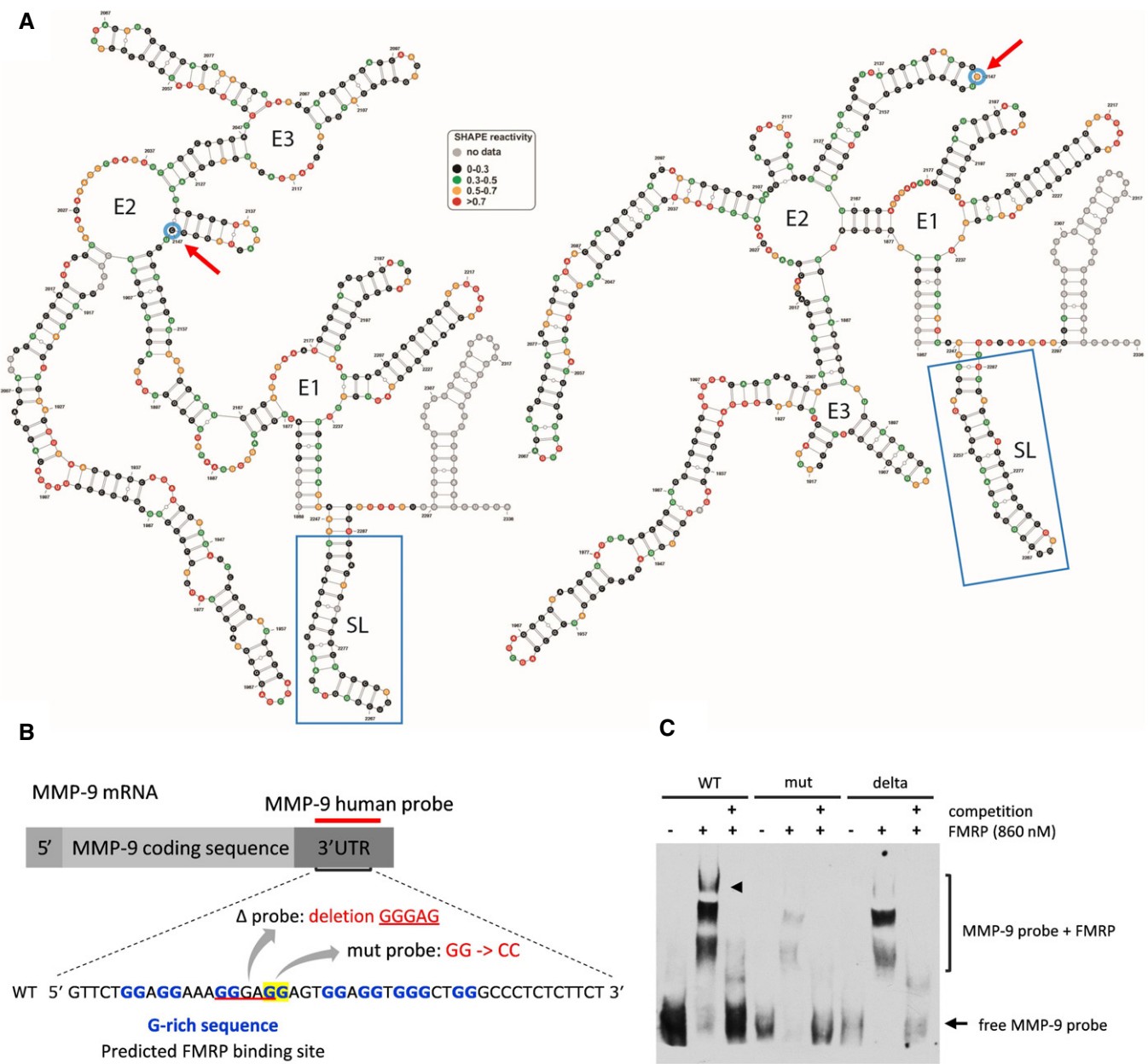

**Figure 2.  Structure of MMP-9 mRNA molecule fragment with either MMP-9_C or MMP-9_T rs20544 polymorphism.**

A   Secondary structure models of 469 nt at the 3′ end of MMP-9 mRNA obtained by RNA structure probing in solution (SHAPE analysis). Blue circles and red arrows, rs20544 polymorphism; blue boxes, G-rich region, predicted FMRP binding site.

B   Sequence of RNA probes that were used in the RNA electrophoretic mobility shift assay (REMSA). The location of the G-rich sequence is indicated.

C   REMSA showing FMRP binding to probes described in (B). The arrowhead indicates the FMRP/MMP-9 RNA complex band that is clearly visible for the human wild-type probe but is hardly detectable for the probe that harbored a mutation or deletion. 100× molar excess of unlabeled probe was added as a competitor to confirm the specificity of the interaction. The figure shows a representative image from three independent experiments.

treatment with forskolin, rolipram, and picrotoxin (Niedringhaus *et al*, 2012; Szepesi *et al*, 2013). cLTP has been shown to increase MMP-9 but not MMP-2 activity (Szepesi *et al*, 2013).

In the colocalization map between the red (RFP) and green (FITC-gelatin) channels, warmer colors correspond to higher gelatinolytic activity (Fig 4A). Interestingly, we found that neurons that were transfected with MMP-9_C had lower MMP-9 activity 15 and 40 min after cLTP induction than observed for MMP-9_T variant (Fig 4B).

### rs20544 SNP influences the morphology of dendritic spines

Synaptic MMP-9 levels can regulate dendritic spine morphology (Vafadari *et al*, 2016). To test whether rs20544 C/T SNP-dependent differences in synaptic MMP-9 levels were followed by changes in dendritic spine morphology, we assessed the relative distribution of spine shapes (mushroom, thin, and stubby) in primary rat hippocampal neurons that overexpressed either the MMP-9_C or

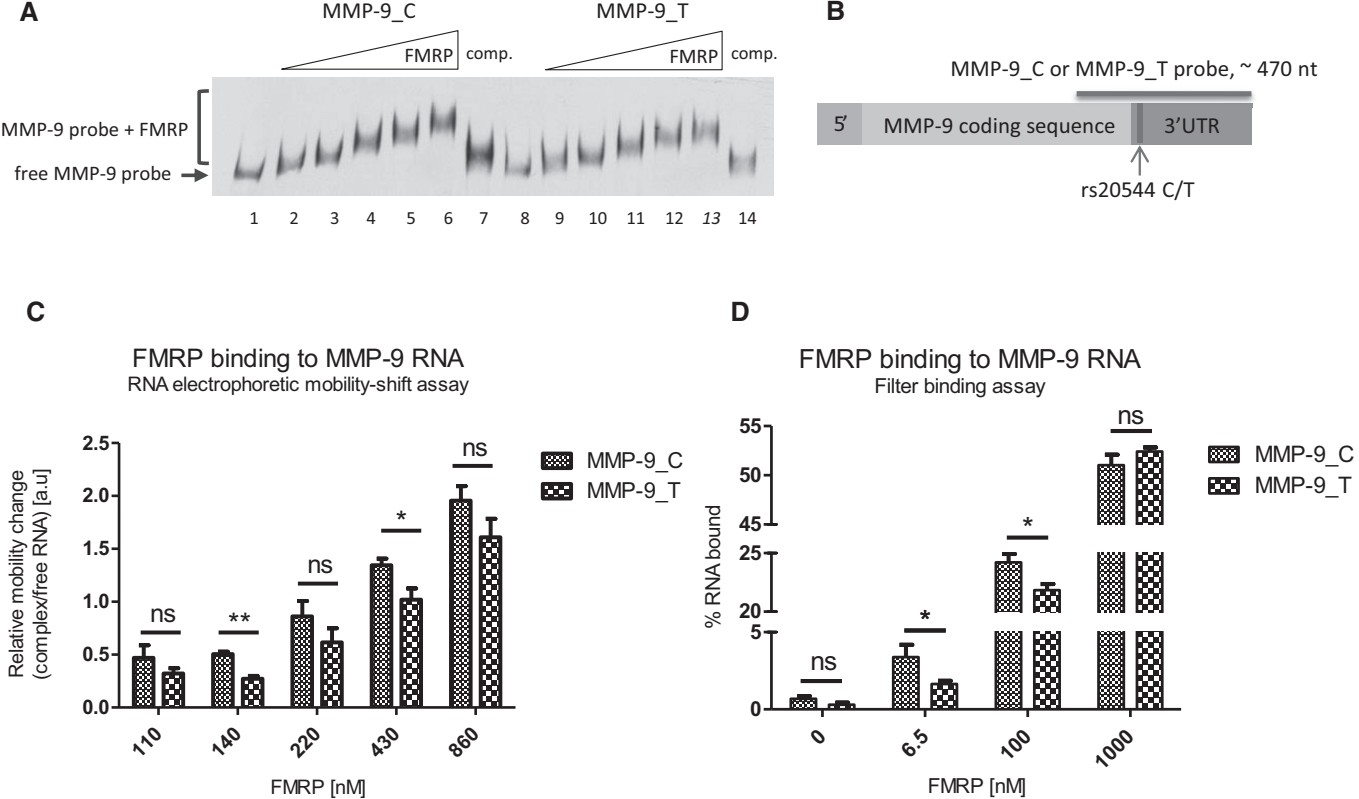

**Figure 3. rs20544 C/T polymorphism affects FMRP binding to MMP-9 RNA molecule.**

A Representative RNA electrophoretic mobility shift assay (REMSA) results. Labeled MMP-9 RNA probe was incubated in the absence (lanes 1 and 8) or presence of increasing amounts of purified FMRP (lanes 2–6 and 9–13). 20× molar excess of unlabeled probe was added as a competitor to confirm the specificity of the interaction (lanes 7 and 14).

B Scheme of MMP-9 mRNA indicating the location of the rs20544 polymorphism and part of the sequence that was used as a probe in the REMSA (A, C) and filter binding assay (D).

C Quantification of REMSA experiments. The relative mobility change of the protein–RNA complexes from the corresponding free probe band was plotted against increasing FMRP concentrations. For each FMRP concentration, the average distance of the shifted complex/free probe band was calculated from at least three independent experiments.

D Quantification of filter binding assay. The fraction of bound RNA was plotted against increasing FMRP concentrations. The data are from five independent experiments. Each column represents the mean counted from range of concentrations (indicated in Materials and Methods), with the final concentration shown on the abscissa.

Data information: Error bars indicate the SEM. *$P < 0.05$, **$P < 0.01$ (Student's *t*-test).

MMP-9_T variant. Figure 4C shows a fragment of a dendrite with higher magnification of representative shapes of mushroom and thin spines.

Time-lapse imaging of living hippocampal neurons revealed significant changes in the distribution of dendritic spine shapes that depended on the *MMP-9* rs20544 C/T SNP. Neurons that overexpressed the MMP-9_C variant had higher percentage of mushroom spines, both in the control conditions and after cLTP induction than neurons that overexpressed MMP-9_T ($P < 0.001$ for all timepoints; Fig 4D). In the control conditions, no difference was observed in the percentage of thin spines for the two SNP variants. However, after cLTP induction, the overexpression of MMP-9_T resulted in a higher percentage of thin spines as compared with the MMP-9_C variant $P < 0.01$ for cLTP$_{15 \text{ min}}$ and $P < 0.001$ for cLTP$_{40 \text{ min}}$; Fig 4D). For both variants, cLTP induction resulted in increased percentage of mushroom spines as compared to the control conditions (MMP-9_C: $P < 0.05$ for cLTP$_{15 \text{ min}}$ and $P < 0.001$ for cLTP$_{40 \text{ min}}$; MMP-9_T:

$P < 0.001$ for cLTP$_{15 \text{ min}}$ and $P < 0.001$ for cLTP$_{40 \text{ min}}$), whereas the stimulation had no effect on thin spines distribution for either variant.

We also analyzed overall spine density, and no difference was found between the SNP variants. The number of protrusions per μm [mean ± SEM] for MMP-9_C was $0.96 \pm 0.05$ for baseline, $1.01 \pm 0.04$ for cLTP$_{15 \text{ min}}$, and $1.06 \pm 0.08$ for cLTP$_{40 \text{ min}}$. The density observed for MMP-9_T variant was $1.03 \pm 0.09$ for baseline, $1.15 \pm 0.09$ for cLTP$_{15 \text{ min}}$, and $1.15 \pm 0.09$ for cLTP$_{40 \text{ min}}$. The induction of cLTP resulted in slight increase in spine density for both variants (for MMP-9_C variant $P < 0.05$ for cLTP$_{40 \text{ min}}$; and for MMP-9_T variant $P < 0.01$ for cLTP$_{15 \text{ min}}$ and cLTP$_{40 \text{ min}}$) (A two-way repeated-measures ANOVA with *post hoc* analysis by Tukey's multiple comparisons).

We then analyzed the main parameters of the mushroom and thin spine categories (i.e., length of thin spines and head area of mushroom spines). Head area has been shown to be proportional to

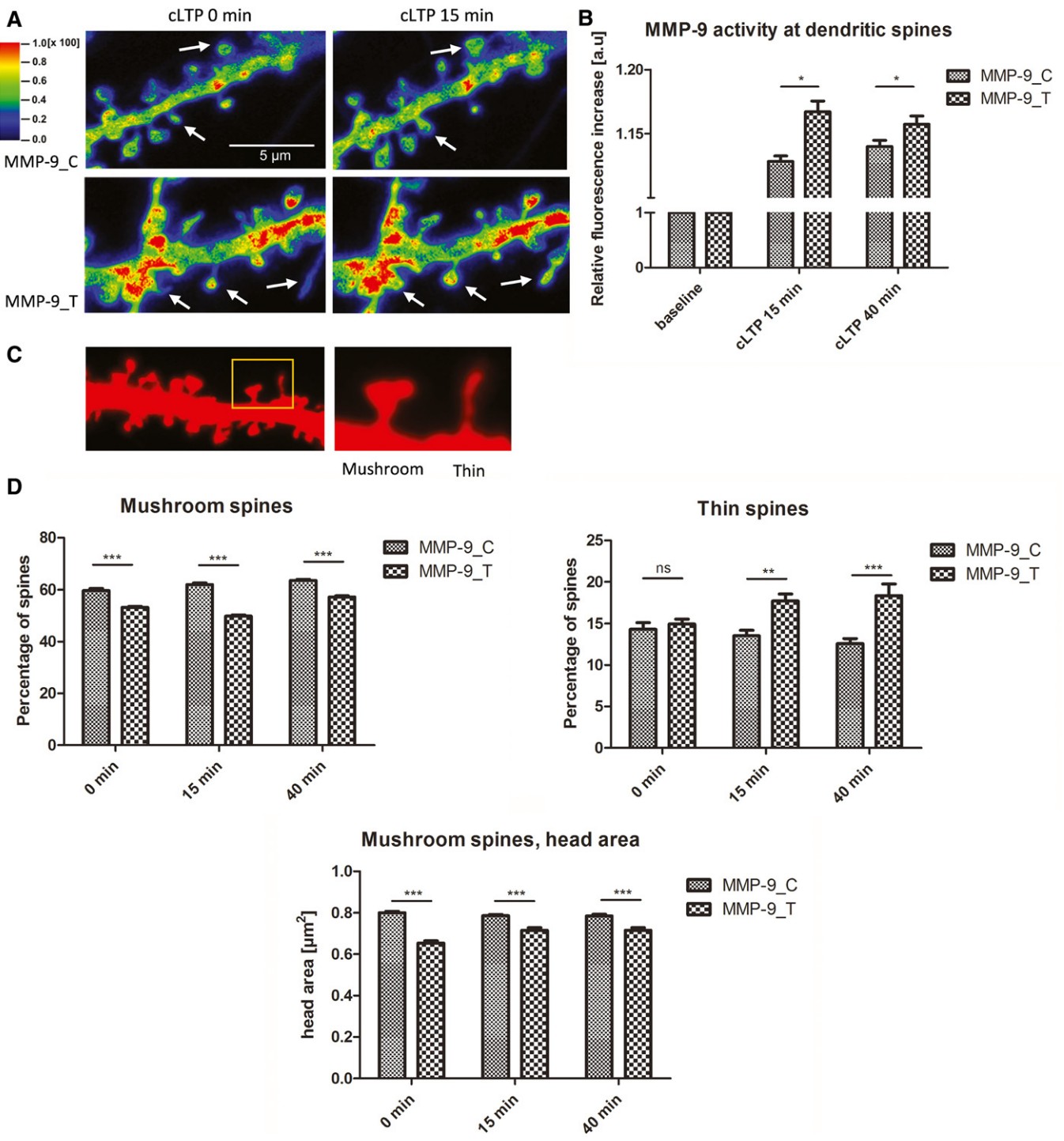

**Figure 4. MMP-9 activity at dendritic spines and analysis of dendritic spine morphology.**

A   Exemplary images from FITC-tagged gelatin (DQ-gelatin) assay showing colocalization map of FITC fluorescence that corresponds to gelatinolytic activity. Red fluorescent protein was used to label cell morphology, showing a short stretch of dendrite that overexpressed either the MMP-9_C or MMP-9_T variant. Warmer colors correspond to higher gelatinolytic activity. Dendritic spines with an increase in gelatinolytic activity are marked with arrows.

B   Quantification of analyzed data from four separate experiments. The relative increase in fluorescence was measured for individual spines over time as a readout of MMP-9 activity. A total of 17 cells (MMP-9_C) or 13 cells (MMP-9_T) were analyzed.

C   Image from morphological analysis of spines from primary rat hippocampal neurons showing example of mushroom and thin spine shapes.

D   The rs20544 polymorphism influenced the percent distribution of spine shape. Quantification of percent distribution of mushroom and thin spines from time-lapse imaging from three separate experiments. Neurons that overexpressed MMP-9_C had more mushroom spines and less thin spines than the MMP-9_T variant. For each polymorphic variant, six cells were analyzed; the total number of spines analyzed was $n_{spines} = 419$ for MMP-9_C and $n_{spines} = 465$ for MMP-9_T.

Data information: Error bars indicate the SEM. *$P < 0.05$, **$P < 0.01$, ***$P < 0.001$ (Mann–Whitney *U*-test in B; two-way ANOVA with multiple comparisons in D).

                        

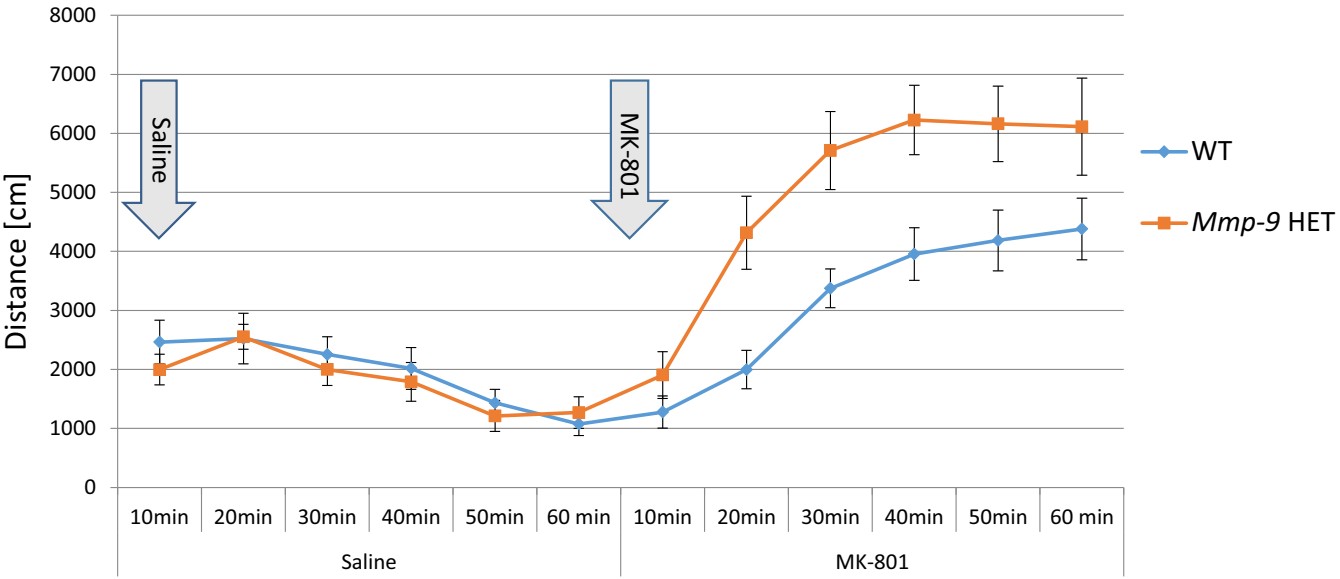

**Figure 5. Psychosis-related locomotor hyperactivity induced by MK-801.**
Decreased levels of active MMP-9 in the brain of *Mmp-9* heterozygous mice result in increased sensitivity to locomotor hyperactivity induced by NMDA receptor antagonist MK-801. After injection of MK-801, heterozygous mice showed significantly greater distance traveled as compared to wild-type mice. $N = 8$ for wild-type mice and $N = 7$ for *Mmp-9* heterozygous mice, data analyzed using two-way repeated-measures ANOVA ($F(1,13) = 5.6163$; $P < 0.05$). HET, heterozygous.

the size of the postsynaptic density and number of presynaptic vesicles (Harris & Stevens, 1989). Neurons that were transfected with MMP-9_C had a significantly larger head area as compared with MMP-9_T transfection ($P < 0.001$ for all timepoints; Fig 4D). We observed increase in head area of mushroom spines for MMP-9_T variant upon cLTP induction ($P < 0.001$ for both $cLTP_{15\ min}$ and $cLTP_{40\ min}$), whereas the stimulation did not affect the head size parameter in neurons transfected with MMP-9_C. Before stimulation, neurons that overexpressed MMP-9_T had longer thin spines ($2.05 \pm 0.07$ μm) compared with MMP-9_C ($2.53 \pm 0.13$ μm; $P < 0.05$). However, after cLTP induction, the difference was no longer observed. For MMP-9_C-overexpressing neurons, cLTP induction caused more pronounced thin spines lengthening than observed for MMP-9_T upon stimulation; therefore, after cLTP induction both variants had similarly long thin spines. In conclusion, neurons that overexpressed MMP-9_C had more mushroom spines, which had a larger head area compared with the MMP-9_T variant.

### *Mmp-9* heterozygous mice exhibit increased sensitivity to MK-801-induced locomotor hyperactivity

Our data for the *MMP-9* rs20544 C/T SNP showed that C carriers (CC/CT) exhibited more severe positive symptoms of schizophrenia than TT carriers. The neuronal culture experiments indicated that the MMP-9_C variant resulted in lower MMP-9 activity after cLTP induction compared with the MMP-9_T variant (Fig 4B). To test whether MMP-9 levels influence performance in a behavioral model of the positive symptoms of schizophrenia, we used *Mmp-9* heterozygous mice with twofold lower levels of MMP-9 in the brain (Fig EV2) and exposed them to psychosis-related locomotor hyperactivity induced by NMDA receptor antagonist MK-801 (van den

Buuse, 2010). Ten minutes after MK-801 administration, *Mmp-9* heterozygous mice exhibited a significant increase in the distance traveled compared with wild-type animals ($F(1,13) = 5.6163$; $P < 0.05$; two-way repeated-measures ANOVA; Fig 5). The saline injection did not cause hyperlocomotion. After the saline treatment, the distance traveled by wild-type and heterozygous animals did not differ significantly. On the other hand, the mice traveled greater distance after MK-801 injection than after the saline injection ($F(9,117) = 46.5515$; $P < 0.01$) and also the treatment effect vs. genotype effect interaction was significantly different ($F(9,117) = 6.1108$; $P < 0.01$). Therefore, lower levels of MMP-9 in the brain in *Mmp-9* heterozygous mice resulted in an increase in MK-801-induced locomotor hyperactivity.

## Discussion

In the present study, we identify a novel function of rs20544 C/T SNP that is located in the 3′UTR of *MMP-9*, and we show that it is strongly associated with the severity of a chronic delusional syndrome in schizophrenia patients. Our case–control results demonstrate that *MMP-9* rs20544 *per se* is not a risk SNP for a schizophrenia endpoint diagnosis, but schizophrenia individuals who carry the C allele (CC/CT) are more severely affected by chronic delusions compared with TT carriers. This analysis was only possible based on a deeply phenotyped sample of schizophrenic individuals, an inevitable prerequisite for approaching the definition of biological disease subgroups. In fact, our results suggest that *MMP-9* contributes to an important biological subgroup under the heterogeneous "umbrella diagnosis" of schizophrenia. These results may also explain why the large GWAS on endpoint diagnoses failed to extract *MMP-9* as a common risk gene for

schizophrenia (Schizophrenia Working Group of the Psychiatric Genomics Consortium, 2014).

Most importantly, we provided mechanistic insights into the way in which the single base change in *MMP-9* rs20544 (C vs. T) remarkably influences gene function. Molecular analyses and structural studies showed that the rs20544 C/T SNP markedly affects the mRNA structure of MMP-9, thus influencing folding of the predicted FMRP binding site (i.e., the G-rich sequence in the 3′UTR). The MMP-9_C variant was bound by FMRP with higher affinity than the MMP-9_T variant. Previous studies showed that MMP-9 mRNA was a subject to activity-driven local translation near excitatory synapses, and FMRP is a potent inhibitor of such translation (Dziembowska *et al*, 2012; Janusz *et al*, 2013; Gkogkas *et al*, 2014). Interestingly, neurons that overexpressed the MMP-9_C variant exhibited lower synaptic MMP-9 activity after cLTP induction compared with neurons that overexpressed the MMP-9_T variant. Moreover, analyses of the morphology of dendritic spines that harbor excitatory synapses confirmed that cLTP-

stimulated neurons overexpressing the MMP-9_T variant were more susceptible to structural plasticity induced by synaptic stimulation, as they had greater abundance of thin spines, which have been previously shown to be more plastic and immature than mushroom spines (Yuste & Bonhoeffer, 2004). A schematic diagram that depicts our proposed model is shown in Fig 6. Consistent with our observations from man to cell is our finding in *Mmp-9* heterozygous mice. These animals exhibited a decrease in MMP-9 levels in the brain and were more sensitive to MK-801-induced locomotor hyperactivity, a model of the positive symptoms of schizophrenia.

RNA structure probing in solution, supported by *in silico* modeling, revealed that the structures of the two variants of MMP-9 mRNA were strikingly different. Such marked changes in the mRNA structure can influence interactions with RNA binding proteins, such as FMRP and short non-coding microRNA molecules. Our previous studies identified FMRP as a potent regulator of the local translation of MMP-9 mRNA (Dziembowska *et al*, 2012; Janusz

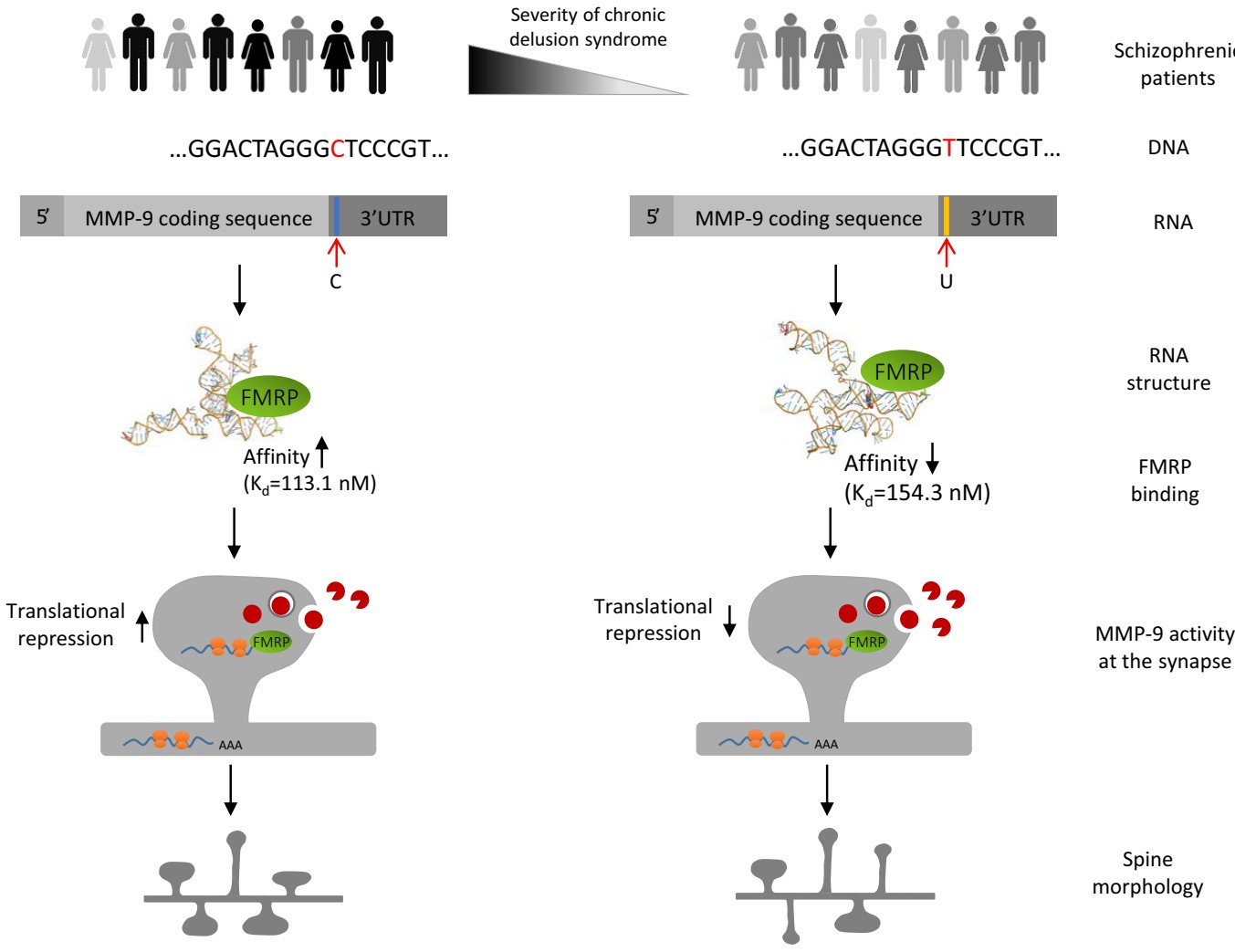

**Figure 6.  Schematic model illustrating the conclusions.**
See text for explanation.

    

*et al*, 2013). The variant-specific structural change in the MMP-9 mRNA molecule was also seen in the region that was predicted to bind FMRP. Although FMRP was previously shown to regulate MMP-9 mRNA (Dziembowska *et al*, 2012; Janusz *et al*, 2013; Gkogkas *et al*, 2014), the exact sequence of the protein–RNA interaction site was not identified. Using a gel-shift assay, we found that FMRP bound MMP-9 mRNA at more than one site. Namely, the ~80 nt human wild-type REMSA probe displayed three protein complex bands (Fig 2C). Two of these binding sites likely correspond to the G-rich sequences that were identified *in silico*.

To assess the binding of FMRP to specific MMP-9 mRNA variants, we used ~470-nt-long probes to ensure the most optimal RNA folding. The observed differences in $K_d$ values were relatively small, possibly reflecting the presence of multiple, yet unidentified binding sites. Additionally, the different secondary and 3D structure organization of the two MMP-9 variants suggests that although both RNA can be bound by FMRP, the complexes that are formed might not be equally functional. Indeed, the REMSA revealed different migration of the FMRP complexes with the MMP-9_T and MMP-9_C probes. Nevertheless, we observed a decrease in FMRP:MMP-9 mRNA complex formation in the presence of 50 mM lithium ions (Fig EV3), suggesting that one of the G-rich sequences that was identified as an FMRP binding site might form a G-quadruplex structure that is destabilized in the presence of lithium ions (Bardin & Leroy, 2008). It has been shown that potassium ions stabilize the G-quartet structures, whereas lithium ions greatly destabilize it (Bardin & Leroy, 2008; Lee *et al*, 2008).

The binding of FMRP to mRNA leads to ribosomal stalling and the inhibition of translation (Darnell *et al*, 2011). Therefore, we expected that its differential affinity for the MMP-9 mRNA variants may result in different synaptic MMP-9 levels. We closely studied the effects of overexpression of either MMP-9_C or MMP-9_T on neuronal activity-induced MMP-9 at dendritic spines and spine morphology. After the induction of cLTP, neurons that overexpressed the MMP-9_T variant, which is bound by FMRP with lower affinity, produced more MMP-9 at the synapse. This can be explained by the inhibitory effect of FMRP on synaptic MMP-9 translation that seems to be more pronounced in the case of the MMP-9_C variant. The higher levels of secreted MMP-9 in MMP-9_T-overexpressing neurons were accompanied by a greater prevalence of thin spines. In contrast, the MMP-9_C variant had more mushroom spines, which are considered more mature and efficient spines. For MMP-9_T transfected neurons, we observed increase in head area of mushroom spines after cLTP induction, whereas stimulation had no effect on this parameter in MMP-9_C-expressing neurons, which had bigger head size than observed for MMP-9_T variant regardless of the cLTP induction. These data are consistent with previous studies (Wang *et al*, 2008; Wilczynski *et al*, 2008; Szepesi *et al*, 2013, 2014; Vafadari *et al*, 2016), arguing that higher MMP-9 levels could confer higher spine plasticity in the MMP-9_T variant, similar to the greater abundance of thin spines and higher MMP-9 activity that are observed in FXS patients and *Fmr1* knockout mice (Rudelli *et al*, 1985; Comery *et al*, 1997; Dziembowska *et al*, 2013). Recently, two studies reported decreased levels of FMRP in brains and blood of subjects with schizophrenia (Fatemi *et al*, 2010; Kovács *et al*, 2013). Our data suggest that different levels of FMRP do not interfere with changes in the affinity observed for the two MMP-9 mRNA variants (Fig 3D). Therefore, we assume that the

alterations in the affinity of FMRP for MMP-9 RNA depending on the SNP, which we show in the present paper, would occur also in a situation of reduced levels of FMRP. Since FMRP negatively regulates MMP-9 mRNA translation and the variant resulting in lower MMP-9 activity at the spines (CC/CT) is the "risk" variant, one could speculate that the downregulation of FMRP in schizophrenia is even a compensatory mechanism to overcome decreased MMP-9 levels.

Locomotor hyperactivity that is produced by NMDA receptor antagonism is frequently used to behaviorally model the positive symptoms of schizophrenia in mice (van den Buuse, 2010), in line with the "hypoglutamatergic hypothesis" of schizophrenia (Gunduz-Bruce, 2009; Radyushkin *et al*, 2010). Despite the obvious shortcomings in addressing uniquely human perceptions, such as hallucinations and delusions, this animal model has advanced the validation of proposed gene mechanisms in schizophrenia. *Mmp-9* heterozygous mice are not the perfect model of the *MMP-9* rs20544 SNP; however, they serve in our study only as supplementary support for the main findings on molecular, morphological, and clinical consequences of the SNP. Locomotor hyperactivity in *Mmp-9* heterozygous mice, which have lower levels of *Mmp-9* expression in the brain, together with decreased MMP-9 activity for the "risk" MMP-9_C variant supports the previously postulated connections between MMP-9 activity, glutamate signaling, and schizophrenia (Lepeta & Kaczmarek, 2015).

In summary, our results link *MMP-9* gene polymorphisms with a specific subphenotype of schizophrenia, suggesting that the underlying mechanism depends on the local synthesis of MMP-9 within the vicinity of excitatory synapses. Synaptic MMP-9, in turn, contributes to the morphology of dendritic spines, which may be viewed as a proxy for synaptic efficacy. Interestingly, another *MMP-9* SNP −1562C/T (rs3918242) has also been shown to affect PANSS score in schizophrenia patients (Ali *et al*, 2017). The present data are also in a good agreement with the results from a recent study that showed that gene risk variants for schizophrenia converge in biological pathways that are involved in excitatory synapses/dendritic spines (Network and Pathway Analysis Subgroup of Psychiatric Genomics Consortium TN and PAS of the PG, 2015). Neuroinflammatory events are another main trait that contributes to the pathophysiology of schizophrenia. Matrix metalloproteinases, owing to their well-known role in inflammatory responses, have been suggested as a novel drug target for the treatment of schizophrenia (Chopra *et al*, 2015). We propose an alternative function for MMP-9 in the pathophysiological underpinnings of schizophrenia, by pointing to a novel mechanism that involves changes in MMP-9 levels at the excitatory synapse and MMP-9-dependent changes in dendritic spine morphology. In conclusion, we found a specific association between a molecular mechanism that involves synaptic MMP-9 and a biological subgroup of schizophrenia, which may open new avenues for individualized treatment of this devastating disease in the future.

# Materials and Methods

## Subjects

### Schizophrenia patients

Göttingen Research Association for Schizophrenia (GRAS) data collection has been ongoing for the past 10 years and consists of

> 1,100 deeply phenotyped patients who were diagnosed with schizophrenia or schizoaffective disorder according to the criteria of the *Diagnostic and Statistical Manual of Mental Disorders*, 4th edition, text revision (American Psychiatric Association, 2000). Patients were recruited from 23 collaborating centers across Germany (Begemann *et al*, 2010; Ribbe *et al*, 2010). The study complies with the Helsinki Declaration and was approved by the Ethics Committee of the Georg-August-University (Göttingen, Germany) and all participating centers. All of the study participants (95.6% European Caucasian, 1.7% other, 2.7% unknown) or, if applicable, their legal representatives provided written informed consent to be included in the study. Of the 1,087 successfully genotyped patients, 66.7% were male (*n* = 725) and 33.3% were female (*n* = 362). Their average age was 39.44 ± 12.56 years (range, 17–79 years).

### Healthy controls

For case–control analyses (Fig 1A), healthy voluntary blood donors were employed. They were recruited in the Department of Transfusion Medicine at the Georg-August-University of Göttingen according to national guidelines for blood donation. As such, they met the necessary health requirements, ensured by a broad pre-donation screening process that included standardized health questionnaires, interviews, and assessments of hemoglobin concentration, blood pressure, pulse, and body temperature. Of the 1,235 successfully genotyped control subjects (97.8% European Caucasian, 2% other, 0.2% unknown), 61.8% were male (*n* = 763) and 38.2% were female (*n* = 472). Their average age was 37.42 ± 13.22 years (range, 18–69 years).

### Genotyping

The GRAS sample was genotyped using a semi-custom Axiom MyDesign Genotyping Array (Affymetrix, Santa Clara, CA, USA), based on a CEU (Caucasian residents of European ancestry from Utah, USA) marker backbone that included 518,722 SNPs and a custom marker set that included 102,537 SNPs. Genotyping was performed by Affymetrix on a GeneTitan platform. Several quality control steps were applied (SNP call rate > 97%, Fisher's linear discriminant > 3.6, heterozygous cluster strength offset > −0.1, and homozygote ratio offset > −0.9) (Hammer *et al*, 2014; Stepniak *et al*, 2015). The genotyping results of the GRAS sample have been included in and published as part of the latest PGC study on schizophrenia (Schizophrenia Working Group of the Psychiatric Genomics Consortium, 2014).

### Phenotyping and target phenotype

Göttingen Research Association for Schizophrenia schizophrenia patients were deeply phenotyped including sociodemographic, psychopathological and other disease-related readouts (Ribbe *et al*, 2010). Basic sociodemographic information, general and schizophrenia-related clinical readouts are presented in Table 1. Screening for a potential association between the *MMP-9* rs20544 SNP and lead symptoms of schizophrenia using the Positive and Negative Syndrome Scale (*PANSS*) (Kay & Fiszbein, 1987) in a phenotype-based genetic association study (PGAS) approach (Hammer *et al*, 2014; Stepniak *et al*, 2014, 2015) pointed to a

chronic delusional syndrome as the relevant phenotype measure. To quantify this target phenotype, a *chronic delusion composite score* was created, based on one item (P1) of the positive subscale of the *PANSS*, two items (N2 and N4) of the negative subscale, and four items (G1, G9, G15, and G16) of the general subscale (Fig 1B). Phenotypic data to construct the *chronic delusion composite score* were available for 1,047 patients.

### Construction of DNA vectors

For the transfection of neurons, the human *MMP-9* gene coding sequence with the 3′UTR was cloned under the synapsin 1 promoter into the pCR4Blunt-TOPO vector (Thermo Scientific), referred to as pCR Syn MMP-9_C. The pCR Syn MMP-9_T construct was obtained by site-directed mutagenesis of the nucleotide that hosted the rs20544 C/T SNP, confirmed by sequencing. For MMP-9_C and MMP-9_T probe preparation for the RNA electrophoretic mobility shift assay, the sequence 469 nucleotides (nt) from the 3′terminus of human MMP-9 mRNA was cloned by polymerase chain reaction (PCR) into KpnI and NotI restriction sites of the pDrive vector, thus creating pDrive REMSA_C and pDrive REMSA_T plasmids. The choice of probe sequence was based on our RNA structure data to maximize the proper RNA structure folding.

### *In silico* MMP-9 mRNA structures

RNA secondary structures were predicted using RNAstructure 5.6 (Deigan *et al*, 2009). The ss-count analysis was performed for the isolated 3′UTRs (192 nt) and full-length (2,336 nt) human MMP-9 RNAs. All of the secondary structures that were predicted within 10% of free energy of the most stable variant were considered, and the window size parameter was changed to generate closely related structures. For the secondary structure prediction that was supported by the experimental data, the normalized reactivity values were introduced to RNAstructure 5.6 software as pseudo-energy constraints with default slope and intercept values. The maximum pairing distance was limited to 600 for the final structures. Differences in nucleotides between groups were analyzed by applying the Wilcoxon rank sum test. The three-dimensional (3D) structures of 469 nt of the 3′-terminal fragment of MMP-9_C and MMP-9_T RNAs were generated using the RNAComposer server (http://rnacomposer.cs.put.poznan.pl; Popenda *et al*, 2012; Biesiada *et al*, 2016). Experimentally supported secondary structures were introduced as an input. The final energy values were used to rank the predicted structures.

### RNA structure probing in solution

Templates for *in vitro* transcription were obtained by the PCR amplification of MMP-9 variants from corresponding plasmids (pCR Syn MMP-9_C and pCR Syn MMP-9_T) using a forward primer that contained a T7 promoter sequence. RNA was transcribed *in vitro* using MEGAscript (Life Technologies) according to the manufacturer's protocol. RNA was treated with DNase I and recovered by LiCl precipitation. For RNA modification, 5 pmol of full-length MMP-9 RNA (MMP-9_C or MMP-9_T variant) was refolded in 100 μl of buffer that contained 10 mM Tris–HCl (pH 8.0), 100 mM KCl, and 0.1 mM ethylenediaminetetraacetic acid (EDTA) with

heating at 95°C for 2 min and then placed on ice. Afterward, 50 µl of 3× folding buffer (120 mM Tris–HCl [pH 8.0], 600 mM KCl, 1.5 mM EDTA, and 15 mM MgCl$_2$) was added and incubated at 37°C for 20 min. In the experiments that were designed to probe the presence of quadruplex structures, the folding buffer contained 120 mM Tris–HCl (pH 8.0), 1.5 mM EDTA, 15 mM MgCl$_2$, and 450 mM LiCl or KCl. Folded RNA was equally divided into two tubes and treated with either 8 µl of 5 mM 1-methyl-7-nitroisatoic anhydride (1m7; reaction) or DMSO alone (control) and allowed to react for 5 min at 37°C. RNA was recovered using the Direct-zol RNA MiniPrep Kit (Zymo Research).

The detection of 2′-*O*-adducts was performed using reverse transcription with fluorescently labeled primers as described previously (Purzycka *et al*, 2013; Nishida *et al*, 2015). Briefly, primers were added to 0.5 pmol of RNA (Cy5 [+] and Cy5.5 [–]; 8 and 6 µM, respectively), and primer-template solutions were incubated at 95°C for 3 min, 37°C for 10 min, and 55°C for 2 min. RNA was then reverse-transcribed at 50°C for 45 min (Invitrogen Superscript III, Life Technologies). Sequencing ladders were prepared using primers that were labeled with WellRed D2 and LicorIR-800 and a Thermo Sequenase Cycle Sequencing kit (Affymetrix) according to the manufacturer's protocol. The samples and sequencing ladders were purified using the ZR DNA Sequencing Clean-up Kit (Zymo Research). Primer extension products were separated using a Beckman CEQ8000 Genetic Analysis System (Beckman-Coulter), and electropherograms were processed using SHAPEfinder software (Vasa *et al*, 2008; Wieczorek *et al*, 2016). *N*-methylisatoic anhydride (NMIA) reactivity was normalized using the ShapeNorm script (Huang *et al*, 2013). Reactivity values were obtained from at least three repetitions and implemented as pseudo-energy constraints to predict RNA secondary structures. In the experiments that sought to detect quadruplexes, the primer extension reactions were performed with [$^{32}$P]-labeled primers as described previously (Purzycka *et al*, 2011).

### Preparation and biotinylation of oligonucleotides

Three types of oligonucleotides were used for the specific experiments:

1    Human wild-type, mutant, and delta MMP-9 RNA probes (~80 nt, sequences listed below) were prepared for REMSA on the template of annealed shorter overlapping oligonucleotides (~50 nt). The annealed product was subjected to PCR-based filling with Taq polymerase (Thermo Scientific) and agarose gel-purified using the Syngen Gel/PCR Mini Kit. The forward primer contained the T7 RNA polymerase promoter sequence to enable *in vitro* transcription of the RNA probe. After DNase I (Roche) treatment, RNA was purified on mini Quick Spin RNA Columns (Roche). RNA integrity was verified by agarose gel electrophoresis. The probes were biotinylated for 2 h at 16°C using the Pierce RNA 3′ End Biotinylation Kit and purified by chloroform extraction. Dot blot assays were performed to assess labeling efficiency.

Wild-type human: 5′-GTTCTGGAGGAAAGGGAGGAGTGGAGG TGGGCTGGGCCCTCTCTTCTCACCTTTGTTTTTTGTTGGAGTG TTTCTAAT-3′

Mutant human: 5′-GTTCTGGAGGAAAGGGACCAGTGGAGGTG GGCTGGGCCCTCTCTTCTCACCTTTGTTTTTTGTTGGAGTGTT TCTAAT-3′

Delta human: 5′-GTTCTGGAGGAAAGAGTGGAGGTGGGCTGGG CCCTCTCTTCTCACCTTTGTTTTTTGTTGGAGTGTTTCTAAT-3′

2    Human MMP-9_C and MMP-9_T probes (~470 nt) for REMSA were obtained by the digestion of pDrive REMSA_C and pDrive REMSA_T with NotI and *in vitro* transcription with RNA polymerase T7 (Roche). Purification and labeling were performed as described above. Biotinylation was conducted overnight.

3    The rat REMSA wild-type probe was prepared by ligating the annealed complementary oligonucleotide that spanned an approximately 50 nt fragment of the rat MMP-9 3′UTR (sequence listed below), including one of the G-rich sequences, into the pCR II vector (TA Cloning Kit, Thermo Scientific). The obtained plasmid was digested with EcoRV enzyme and subjected to *in vitro* transcription with RNA polymerase Sp6 (Roche). In addition to the MMP-9 3′UTR fragment, the probe also contained an approximately 75 nt sequence that was present in the vector between the cloning site and Sp6 promoter. The probe was purified and biotinylated as described above. This probe was used to choose the optimal FMRP concentration range for further experiments with human probes. The titration of increasing amounts of FMRP caused the appearance of a shifted band that indicated complex formation (Fig EV4). The specificity of the interaction was confirmed by competition with an excess of unlabeled probe.

Wild-type rat: 5′-TGGTGATCTCTTCTAGAGACTAGGAAGGA GTGGAGGCGGGCAGGGCCCTCTCTGCA-3′

### FMRP purification

The rat *Fmr1* nucleotide sequence was cloned into the pET28 vector using a sequence- and ligation-independent cloning (SLIC) method. The plasmid contained an N-terminal His6-tagged SUMO protein sequence that can be enzymatically cleaved by SUMO protease. The plasmid was propagated in *E. coli* BL21-CodonPlus-RIL and purified by the ÄKTA™xpress chromatography system at 10°C. Briefly, after centrifugation, the supernatant was loaded on a Ni-NTA Superflow Agarose column (Qiagen). Bound protein was on-column cleaved by SUMO protease (20 µg protease per 5 ml of resin) in elution buffer (20 mM Tris [pH 8], 600 mM NaCl, and 300 mM imidazole) for 8 h at 10°C. The cleaved protein was further purified by combinations of a HiTrap desalting column (GE Healthcare) and Ni-NTA Agarose column. Protein that was not bound with resin was collected and automatically loaded onto a Superdex 200 column (GE Healthcare) that was pre-equilibrated with GF buffer (10 mM Tris [pH 8] and 600 mM NaCl). After lowering the salt concentration to 100 mM NaCl by adding four volumes of 10 mM Tris (pH 8), the proteins were further purified by ion exchange chromatography on a ResourceQ column (GE Healthcare) that was equilibrated with LS buffer (10 mM Tris [pH 8] and 100 mM NaCl) and eluted with HS buffer (10 mM Tris [pH 8] and 1 M NaCl). Purified proteins were analyzed by 12% sodium dodecyl sulfate–polyacrylamide gel electrophoresis (SDS–PAGE).

### RNA electrophoretic mobility shift assay

RNA electrophoretic mobility shift assay was performed according to the Pierce kit's instructions (LightShift Chemiluminescent RNA

EMSA Kit, Thermo Scientific), with modifications. Biotinylated RNA probes were denatured for 5 min at 95°C and allowed to cool for 5 min at room temperature. RNA (10–20 fmoles) was incubated with purified full-length FMRP (concentrations indicated in figures). Binding was performed at room temperature for 20–30 min in custom-made binding buffer (150 mM KCl, 0.5 mM DTT, 20 mM HEPES [pH 7.6], 5% glycerol, and 50 ng/μl tRNA). In some of the experiments, a 20–200× molar excess of unlabeled probe was added as a competitor to confirm the specificity of the interaction (amounts indicated in legends of Figs 2C and 3A, and EV4). After the addition of 4 μl of 6× DNA Loading Dye (Thermo Scientific), the mixture was loaded on 5% native polyacrylamide gel (duracryl, 0.5% glycerol, and 0.5× TBE).

Electrophoresis was performed in 0.5× TBE buffer (pH ~8.3) for 2–3 h at 170 V and 4°C. After transfer, the nylon membrane was ultraviolet-cross-linked at 120 mJ/cm$^2$ and developed according to the Pierce kit's instructions (LightShift Chemiluminescent RNA EMSA Kit, Thermo Scientific). For the MMP-9_C and MMP-9_T probes, the relative mobility change of the protein–RNA complexes was calculated as the distance from the corresponding free probe band. For quantification, the relative mobility change was plotted against increasing FMRP concentrations. For each FMRP concentration, the average distance of the shifted complex/free probe band was calculated from at least three independent experiments (110–220 nM FMRP, from three independent experiments; 430 nM FMRP, from nine independent experiments; 860 nM FMRP, from eight independent experiments).

### Filter binding assay

The experiments were performed as described (Nishida *et al*, 2015), with modifications. RNA was labeled at the 3′ end using T4 RNA ligase (Thermo Scientific) and [$^{32}$P]pCp overnight at 4°C and column-purified with the Direct-zol RNA MiniPrep Kit (Zymo Research). Radiolabeled RNA probes (20 fmoles) were denatured for 2 min at 95°C and folded at 37°C for 10 min in binding buffer (10 mM Tris [pH 7.5], 50 mM KCl, and 1 mM DTT; Thermo Scientific). Renaturated RNA was incubated with increasing amounts of full-length FMRP in a final volume of 20 μl for 20–30 min at room temperature and then filtered through wet nitrocellulose membranes (Protran BA79 and Amersham Hybond-N$^+$) under gentle suction. After filtration, the membranes were washed twice with 200 μl of wash buffer (10 mM Tris [pH 7.5] and 50 mM KCl), dried, and exposed to a phosphoimager screen. The data were analyzed using ImageJ and GraphPad Prism 6 software. The filter binding experiments were repeated five times (MMP-9_C and MMP-9_T probes) or three times (wild-type, mutant, and delta probes).

### Neuronal culture and transfection

Dissociated hippocampal cultures were prepared from postnatal day 0 (P0) wild-type Wistar rats (pups of either sex) as described previously (Jasińska *et al*, 2015). The neuronal cultures were co-transfected after 10 days of culture *in vitro* (DIV10) with vectors for the overexpression of RFP together with MMP-9_C or MMP-9_T for verification of transfection efficacy using Lipofectamine 2000 (Life Technologies), according to manufacturer's protocol. In the pilot

experiment, zymography was performed to verify overexpression of MMP-9 in culture transfected with MMP-9-C or MMP-9-T vectors comparing the levels of MMP-9 to control culture transfected only with RFP coding vector.

### Cell culture stimulation and gelatinase assay in living neuronal culture

The experiments that are described below were performed on DIV19-23. The cells were first incubated in maintenance medium with a mixture of 1 μM tetrodotoxin (TTX), 40 μM 6-cyano-7-nitro-quinoxaline-2,3-dione (CNQX), 100 μM (2*R*)-amino-5-phosphonovaleric acid (APV), and 5 μM nimodipine (all from Sigma-Aldrich) for 3 h to silence spontaneous neuronal activity. To visualize gelatinase activity, fluorescein conjugate gelatin (DQ-gelatin; Molecular Probes, catalog no. D12054) was added to the cells 30 min before the end of silencing at a final concentration of 40 ng/μl as described previously (Szepesi *et al*, 2014). The fluorescence of DQ-gelatin is quenched until it is digested by gelatinases, MMP-9, and MMP-2 in neuronal cultures. Therefore, an increase in fluorescence is proportional to the proteolytic activity of gelatinases. At the end of silencing, the cells were briefly washed and kept in conditioned culture medium that contained 1 μM TTX, 40 μM CNQX, 5 μM nimodipine, and a mixture of 50 μM forskolin, 0.1 μM rolipram, and 50 μM picrotoxin (all from Sigma-Aldrich), which were previously shown to induce chemical long-term potentiation (cLTP) *in vitro* (Otmakhov *et al*, 2004; Szepesi *et al*, 2013).

### Live cell imaging

Live cell imaging was performed with 3-week-old cultured hippocampal neurons in a living chamber at 37°C in a 5% CO$_2$ atmosphere as described previously (Szepesi *et al*, 2013; Szepesi *et al*, 2014). As a readout of MMP-9 activity, the level of fluorescence from digested DQ-gelatin was measured for individual spines using time-lapse imaging 0, 15, and 40 min after cLTP induction. Dendritic spines from randomly selected segments of secondary or tertiary dendrites were imaged in four independent experiments ($n = 17$ cells for MMP-9_C and $n = 13$ cells for MMP-9_T; an average of 188 spines/cell for MMP-9_C and 147 spines/cell for MMP-9_C). Only dendritic spines that persisted during the entire imaging session were analyzed ($n_{spines} = 3,197$ for MMP-9_C and $n_{spines} = 1,906$ for MMP-9_T). Of these spines, only those that exhibited an increase in fluorescence over time were used for further quantification ($n_{spines\ cLTP\ 15\ min} = 1,134$ and $n_{spines\ cLTP\ 40\ min} = 1,216$ for MMP-9_C, constituting 35.47% and 38.04% of total spines, respectively; $n_{spines\ cLTP\ 15\ min} = 761$ and $n_{spines\ cLTP\ 40\ min} = 736$ for MMP-9_T, constituting 39.93% and 38.61% of total spines, respectively). Images were acquired using a Zeiss LSM780 confocal microscope with a 40× water-immersion objective and 488 and 561 nm laser lines at 1,024 × 1,024 pixel resolution. A series of 10 *z*-stacks was acquired for each cell at 0.42 μm steps, with additional digital zoom at final lateral resolution of 0.07 μm per pixel.

Images were further processed using ImageJ software (National Institutes of Health). The RFP signal (red channel) was used to create a mask that was applied to DQ-gelatin images (green channel) to enable the quantification of only transfected neurons that

overexpressed either the MMP-9_C or MMP-9_T variant. The signal was enhanced for all images by multiplication values of 1.2 for the red channel and 3 for the green channel. The $x$–$y$ cell drift over time was corrected using the StackReg ImageJ plugin (Thévenaz *et al*, 1998). Because DQ-gelatin was washed off with changes in the medium for wash and stimulation, only gelatin that adhered to the cells remained. As a result, the overall green channel signal decreased and was corrected accordingly.

## Dendritic spine analysis

For the analysis of dendritic spine morphology, images from the DQ-gelatin assay that was performed on primary rat hippocampal neurons were used (see above). Using semi-automated custom-made SpineMagick! Software (Ruszczycki *et al*, 2012), the spine area, length, and width were measured. The spines were then clustered using custom scripts that were written in Python with the NumPy, SciPy, and Matplotlib into three main categories as described previously (Jasińska *et al*, 2015): spines with a clear head and neck (mushroom spine), filopodia-like spines (thin spines), and spines without heads (stubby spines). Clustering was further manually corrected based on average images of the clusters and visual inspection of spines that comprised the clusters. The percentage of distribution of the spine shapes was calculated for neurons that overexpressed either the MMP-9_C or MMP-9_T variant. The overall spine density was measured as a number of protrusions per μm and reported as mean ± SEM. For each polymorphic variant, six cells from three separate experiments were analyzed. The total number of spines analyzed was $n_{spines} = 419$ for MMP-9_C and $n_{spines} = 465$ for MMP-9_T.

## Gel zymography

The cortex and hippocampi were dissected from heterozygous MMP-9$^{+/-}$ mice and control wild-type (MMP-9$^{+/+}$) mice. Tissue was homogenized and divided into Triton-soluble and Triton-insoluble (enriched in MMP-9) fractions as described previously (Szklarczyk *et al*, 2002). The protein concentration in the Triton-insoluble fraction was measured using the BCA kit (Pierce). A total of 50 μg protein was loaded per lane. The samples were mixed with 5× Laemmli sample buffer and loaded onto 8% acrylamide gels that contained 2 mg/ml FITC-gelatin (custom made). After SDS–PAGE, the gels were washed twice for 20 min in 2.5% Triton X-100 and incubated for 10 days in zymography buffer (50 mM Tris [pH 7.5], 10 mM CaCl$_2$, 1 μM ZnCl$_2$, and 1% Triton X-100) at 37°C with gentle rocking. The gels were visualized under ultraviolet light. The intensity of white bands against the dark background was quantified using ImageJ software. MMP-9 activity was normalized to MMP-2 activity.

## Locomotor hyperactivity

The experiments were performed on 2- to 3-month-old C57BL/6J *Mmp-9* heterozygous mice and their wild-type siblings. Strain colony was maintained in the animal house of the Nencki Institute. Mice were maintained individually in ventilated cages with dimensions of 38 cm (length) × 23 cm (width) × 20 cm (height) with continuous access to water and food. The animals were provided with nesting

### The paper explained

**Problem**

Schizophrenia has a strong genetic component, but it is unclear how genetic variants contribute to the disease phenotype. Synaptic plasticity underlies reorganization of neuronal circuitry and has been regarded as playing a major role in schizophrenia. A link between a key regulator of dendritic spines' (small dendritic protrusions that harbor excitatory synapses) plasticity, MMP-9 (matrix metalloproteinase), and schizophrenia has been postulated, but no explanation of the underlying molecular mechanisms has been previously provided.

**Results**

We report here that *MMP-9* 3′UTR gene polymorphism does not increase the general risk of schizophrenia, but it contributes significantly to the chronic delusional syndrome in schizophrenia patients. To reveal the underlying mechanism, we employ *in silico* modeling of MMP-9 RNA structure, and next we probe and confirm the results with biochemical experiments, discovering direct binding of Fragile X mental retardation protein (FMRP, an RNA binding protein) to MMP-9 mRNA. Next, we demonstrate that because of the different efficacy of the MMP-9 mRNA polymorphic variants binding to FMRP, there are different levels of synaptic MMP-9 activity, which is locally produced and released to regulate dendritic spine morphology. The chain of interactions thus reads: C allele of this *MMP-9* gene polymorphism—higher affinity of MMP-9 mRNA-FMRP binding—less MMP-9 produced at the excitatory synapses—altered morphology of dendritic spines—more severe chronic delusions.

**Impact**

In conclusion, we found a specific association between a molecular mechanism that involves synaptic MMP-9 and a biological subgroup of schizophrenia, which may open new avenues for individualized treatment of this devastating disease.

material. The following conditions were provided for the mice: temperature of 21–23°C, humidity 50–60%, daily cycle 12/12 h (light phase from 7 to 19). One animal was excluded because of partially missing data. Seven *Mmp-9* heterozygous mice (four females and three males) and eight wild-type mice (four females and four males) were subjected to the experiment.

Locomotor hyperactivity was induced by the noncompetitive NMDA receptor antagonist MK-801 (Sigma-Aldrich) as described (van den Buuse, 2010), with modifications. Mice were habituated to handling by the experimenter and the experimental room. The mice were then individually transported to the experimental room (with blinding to the genotype), placed in an open-field apparatus (40 cm × 40 cm), and allowed to freely explore the apparatus for 30 min. The mice then received an intraperitoneal injection of physiological saline and were returned to the open field for 1 h. Finally, the animals received an intraperitoneal injection of 0.25 mg/kg MK-801 and were allowed to explore the apparatus for 1 h. Behavior was monitored by a video camera that was placed above the center of the apparatus. The data were analyzed using EthoVision XT (Noldus). The distance traveled was analyzed starting 10 min after saline or drug injection.

All of the procedures were performed in accordance with the Animal Protection Act of Poland (Directive 2010/63/EU) and were approved by the 1$^{st}$ local ethics committee (permission no. 257/2012).

## Statistical analysis

Sample sizes were designed to give statistical power. The case–control analysis and tests for deviation from Hardy–Weinberg Equilibrium (HWE) were performed using PLINK 1.07 software. The statistical analyses of phenotype–genotype associations in the human samples were performed using SPSS 17.0 software. For the analysis of the *chronic delusion composite score*, the Spearman rank correlation coefficient was calculated to assess the strength of association between two nonparametric variables. Cronbach's α was calculated as a measure of internal consistency. In the present study, carriers of the *MMP-9* rs20544-C allele (risk allele; CC/CT) were compared with TT carriers. Genotype differences with regard to the clinical measures were tested nonparametrically using the Mann–Whitney *U*-test (continuous variables).

Two-tailed Student's *t*-test was used to analyze filter binding and data from the RNA gel-shift experiments. For the morphological analysis of dendritic spines, two-way analysis of variance (ANOVA) with *post hoc* analysis by Tukey's multiple comparisons was used. Spine density was measured using two-way repeated-measures ANOVA with *post hoc* analysis by Tukey's multiple comparisons. The Mann–Whitney *U*-test was used for the DQ-gelatin assay. For the behavioral experiment, two different groups were analyzed: wild-type and heterozygous after injection of saline & MK-801 by two-way repeated-measures ANOVA.

The statistical analyses were performed using GraphPad Prism 6 & STATISTICA 8 software. The data are expressed as mean ± standard error of the mean (SEM) except for the Fig EV3, on which mean ± standard deviation (SD) is shown. Values of $P < 0.05$ were considered statistically significant. The exact *P*-values from all experiments described in the manuscript are reported in the Table EV1.

**Expanded View** for this article is available online.

## Acknowledgements

The authors would like to thank Michael Arends for carefully editing the manuscript. This work was supported by The Foundation for Polish Science (FNP) TEAM grant, the EXTRABRAIN 7th Framework Marie Curie European Union Initial Training Network grant and auxiliary support from the Polish Ministry of Science and Higher Education (UMOWA Nr 2948-7.PR-2013-2). GRAS was supported by the Max Planck Society, the Max Planck Förderstiftung, the DFG (CNMPB), EXTRABRAIN EU-FP7, the Niedersachsen-Research Network on Neuroinfectiology (N-RENNT), and EU-AIMS.

## Author contributions

Conceived the study: HE, MD, LK, and RWA Designed, performed, and analyzed the experiments: KL, KJP, KP-W, MM, MB, and BV Contributed unpublished reagent (purified FMRP): KB Wrote the manuscript: KL, MD, HE, and LK.

## Conflict of interest

The authors declare that they have no conflict of interest.

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
