## [Review Process File · EMBO Molecular Medicine]

A normal genetic variation modulates synaptic MMP-9 protein levels and the severity of schizophrenia symptoms

Katarzyna Lepeta, Katarzyna J. Purzycka, Katarzyna Pachulska-Wieczorek, Marina Mitjans, Martin Begemann, Behnam Vafadari, Krystian Bijata, Ryszard W. Adamiak, Hannelore Ehrenreich, Magdalena Dziembowska, and Leszek Kaczmarek

Corresponding author: Leszek Kaczmarek, Nencki Institute

Review timeline:

Submission date:	21 February 2017
Editorial Decision:	04 March 2017
Revision received:	22 April 2017
Editorial Decision:	09 May 2017
Revision received:	15 May 2017
Accepted:	15 May 2017

Transaction Report:

Editor: Céline Carret

1st Editorial Decision

04 March 2017

Thank you for the submission of your manuscript to EMBO Molecular Medicine and for your patience while the manuscript was being peer-reviewed. We have now heard back from the three referees whom we asked to evaluate your manuscript.

As you will see from the comments below, the three referees are enthusiastic about the study and do have suggestions and recommendations to further improve conclusiveness and clarity.

We would like to give you the opportunity to revise your manuscript, with the understanding that the referees concerns must be fully addressed. Please note EMBO Molecular Medicine encourages one single round of revision and that, as acceptance or rejection of the manuscript may depend on another round of review, your responses should be as complete as possible.

I look forward to receiving your revised manuscript.

***** Reviewer's comments *****

Referee #1 (Remarks):

This is an interesting study that discusses the apparent impact of a MMP-9 variant in schizophrenic subjects on severity of delusions. The following issues need to be clarified: 1-the generation of a delusional subscore which has not been verified by prior literature is of concern as it is based on a mixture of both positive and negative symptoms and can no longer be called delusions, 2-the details of demographic data such as medication history, handedness, history of drug or cigarette use etc should be discussed, 3-As levels of FMRP in brains and blood of subjects with schizophrenia maybe lower, authors need to discuss the impact of this factor on binding of MMP-9

Referee #2 (Remarks):

Employing the deep-phenotype data of schizophrenic patients from their GWAS previously published, the authors report a significant association between chronic delusion in these patients and a SNP in 3'untranslated region of MMP-9. Following up on this finding, they next performed and report a remarkably thorough investigation into the structural, morphological and functional consequences of this polymorphism. Based on their results, the authors propose that MMP-9 modulates the severity of schizophrenia symptoms by its influence on the morphology of dendritic spines causing a decrease in the efficacy of excitatory NMDA synapses. Finally, the authors employed an animal model of schizophrenia to test their "hypoglutaminergic hypothesis". The results were in line with the predictions.

Suggested minor changes: The importance of deep phenotyping should be addressed.

Recommendation: The authors present novel and important findings that deserve to be published in EMBO Molecular Medicine. I recommend the publication without any hesitation.

Referee #3 (Remarks):

In the studies authors identified new SNP in 3'UTR of MMP9 mRNA. Although there was no significant difference in occurrence of CC/CT polymorphic variants between schizophrenia patients and control subjects, a significant correlation was noted between a chronic delusion composite score and occurrence of this SNP in schizophrenia patients. The findings are very interesting and suggest that reduced synaptic levels of MMP9 may exacerbate the behaviors in schizophrenia patients. While MMP9 was targeted for treatment of schizophrenia and increased plasma levels of MMP9 were reported in the patients, current studies suggest that decreased levels of MMP9 in synapses may contribute to behaviors associated with schizophrenia. Overall experiments are executed well. However, there are some minor concerns regarding analysis and presentation of the results. The paper is suitable for publication with minor revisions.

Minor concerns:

1. How MMP9-C and MMP9-T overexpression levels were quantified in the transfected hippocampal neuron cultures?
2. Was the density of mushroom spines also lower in MMP-T expressing neurons or only percent? Was overall spine density different between MMP9-C and MMP9-T expressing neurons?
3. Did two-way ANOVA show differences in the proportion of mushroom/thin spines or spine head size at 15 or 40 min after cLTP induction as compared to 0 min in neurons expressing MMP9-T or

MMP9-C (fig4)? This should be discussed.

4. Fig S3 shows only pro-forms of MMP9 and MMP2, but figure legends indicates that it shows "levels of active MMP-9" This should be clarified.

5. Two-way ANOVA should be used for statistical analysis of mouse behavior (genotype, WT versus Het and treatment condition, saline versus MK-801). Same mice were tested in the open field before and after injections of NMDAR antagonists. Was this taking into consideration during statistical analysis?

1st Revision - authors' response

22 April 2017

Please find enclosed a revised version of our manuscript entitled "A normal genetic variation modulates synaptic MMP-9 protein levels and the severity of schizophrenia symptoms". We thank Reviewers for their comments and insightful questions. We have tried to address all of the comments as carefully as we could.

Referee #1 (Remarks):

1. The generation of a delusional subscore which has not been verified by prior literature is of concern as it is based on a mixture of both positive and negative symptoms and can no longer be called delusions,

We agree with the reviewer that the **chronic delusion composite score** is new as subscore of the well-established PANSS, but we still feel that it most adequately reflects the behavioral phenotype that is influenced by the risk genotype (C allele). It would certainly have been easy for us to just use the PANSS P1 item which appears to take the 'lead' as far as significance is concerned ($p=0.0005$). See Figure 1D. Adding the other (borderline significant) PANSS signals to the composite results in 'only' $p=0.0003$. Nevertheless, together with the other PANSS signals, the **clinical picture of risk allele carriers** is much better and more adequately described. PANSS P1 alone does not distinguish between acute or chronic delusion. The clinical picture of C carriers, however, is associated with social and emotional withdrawal, somatic concern, unusual thought content and preoccupation which together form the composite score used here for our PGAS study. This is presented in a completely transparent fashion. The reader can clearly see not only the score composition (Figure 1B), but also the relative statistical weight of its parts (Figure 1D). Since this is the overall result we obtained, we would not really want to skip any of them. We now slightly modified the respective paragraph page 7/8.

2. The details of demographic data such as medication history, handedness, history of drug or cigarette use etc should be discussed,

We have now added a table on the requested information (sociodemographic and disease-related parameters) which will be added to the manuscript. Please note that there are no significant differences between risk allele (CC/CT) and TT carriers. The Results and Materials & Methods sections have been updated accordingly (page 7/8 and 21/22, respectively).

3. As levels of FMRP in brains and blood of subjects with schizophrenia maybe lower, authors need to discuss the impact of this factor on binding of MMP-9

We thank the Reviewer for the very interesting comment. The findings of lower levels of FMRP in the brain and blood of subjects with schizophrenia have now been discussed on page 18.

Referee #2 (Remarks):

Suggested minor changes:

The importance of deep phenotyping should be addressed.

We thank the Reviewer for the very positive and stimulating feedback. The importance of deep phenotyping has now been further addressed, see modified paragraph in the discussion, page 15.

Referee #3 (Remarks):

Minor concerns:

1. How MMP9-C and MMP9-T overexpression levels were quantified in the transfected hippocampal neuron cultures?

As suggested by the Reviewer, the appropriate description of the procedures has been added to the revised manuscript in Materials & Methods, on page 28.

2. Was the density of mushroom spines also lower in MMP-T expressing neurons or only percent? Was overall spine density different between MMP9-C and MMP9-T expressing neurons?

To comply with this request, the results from the spine density analysis have been added to the revised manuscript. The Results (page 12) and Materials & Methods (page 31) sections have been updated accordingly.

3. Did two-way ANOVA show differences in the proportion of mushroom/thin spines or spine head size at 15 or 40 min after cLTP induction as compared to 0 min in neurons expressing MMP9-T or MMP9-C (fig4)? This should be discussed.

A two-way ANOVA with post-hoc analysis by Tukey's multiple comparisons was performed to assess the effect of polymorphic variant and cLTP stimulation on the proportion of spine types as well as on the mushroom spines' head size. To comply with the Reviewer's request, the results on the effect of stimulation for each polymorphic variant have been added to the revised manuscript in the Results section (page 12/13) and in the Discussion, on page 17.

4. Fig S3 shows only pro-forms of MMP9 and MMP2, but figure legends indicates that it shows "levels of active MMP-9" This should be clarified.

Please note that due to the reformatting of the manuscript, the order and naming of Supplementary Figures has been changed. Fig S3 is now Fig. EV2 with figure legend in the Supplementary Information, page 1, which has been corrected appropriately.

5. Two-way ANOVA should be used for statistical analysis of mouse behavior (genotype, WT versus Het and treatment condition, saline versus MK-801). Same mice were tested in the open field before and after injections of NMDAR antagonists. Was this taking into consideration during statistical analysis?

To comply to this request, results from two-way repeated measures ANOVA have been added to the revised manuscript in the Results section, page 14 and in Materials & Methods, on page 33.

We appreciate your careful evaluation of our work that helped us to improve the quality of the paper. We hope that this revision meets with your approval. We have included the revised manuscript version that highlights in yellow the changes from the original submission.

Thank you again for your interest in our work. We await your review of our revised manuscript.

2nd Editorial Decision

09 May 2017

Thank you for the submission of your revised manuscript to EMBO Molecular Medicine. We have now received the enclosed reports from the referees that were asked to re-assess it. As you will see the reviewers are now supportive and I am pleased to inform you that we will be able to accept your manuscript pending the following final editorial amendments:

1) Animal ethics details: in the paragraph about neuronal culture of rats brains, you have provided no details about the animals. Please do so: age, gender, background (wild type?) etc.

2) Genotype deposition: we duly note that you did not obtain explicit consent to deposit the clinical data into EGA (or else). However I am afraid that you must do so. EGA allows strict access control of datasets should you need it. Please see below:

It is possible to submit information to the EGA while still continuing to manage access via a Data Access Committee (DAC): <https://www.ebi.ac.uk/ega/home>

It's important to stress that the Data Access Committee - which one would need to allow access to the raw data in some way - would remain unchanged. Many studies, each with managed access, do this (see: <https://www.ebi.ac.uk/ega/datasets>).

Be warned that it often takes quite a bit of time for submission. This can be fast tracked but more like 3 or 4 weeks rather than 3 or 4 days. This is because one's data access committee needs to be set up, documentation submitted around it, etc.

Please think of updating the author's checklist as you get the accession number.

TEXT FROM EGA:

Who controls access to this dataset?

For each dataset that requires access control, there is a corresponding Data Access Committee (DAC) who determines access permissions. Data access requests are reviewed by the relevant DAC, not by the EGA.

The text within the study could look like this: "Our datasets were obtained from subjects who have consented to the use of their individual genetic data for biomedical research, but not for unlimited public data release. Therefore, we submitted it to the European Genome-phenome Archive, through which researchers can apply for access of the raw data."

I look forward to reading a new revised version of your manuscript as soon as possible.

***** Reviewer's comments *****

Referee #1 (Remarks):

The revision is acceptable to me. I recommend publication.

Referee #3 (Remarks):

Revisions have addressed the reviewer's concerns and the paper is appropriate for publication in its present form.

2nd Revision - authors' response

15 May 2017

Authors made the requested editorial changes.

Corresponding Author Name: Leszek Kaczmarek
 Journal Submitted to: EMBO Molecular Medicine
 Manuscript Number: EMM-2017-07723-V2